# Effect of Nail Grips on Weight Bearing and Limb Function in 30 Dogs 2 Weeks Post Tibial Plateau Leveling Osteotomy

**DOI:** 10.3390/ani12182312

**Published:** 2022-09-06

**Authors:** Jennifer Repac, Leilani X. Alvarez, Kenneth E. Lamb, Daniel Spector

**Affiliations:** 1College of Veterinary Medicine, University of Florida, Gainesville, FL 32608, USA; 2The Animal Medical Center, New York, NY 10065, USA; 3Lamb Statistical Consulting and Scientific Writing LLC, West Saint Paul, MN 55118, USA

**Keywords:** canine, TPLO, assistive device, nail grips, tibial plateau leveling osteotomy

## Abstract

**Simple Summary:**

Devices made to improve mobility are becoming increasingly popular among owners of dogs with orthopedic conditions, but there are few studies investigating their efficacy. Nail grips are commonly used by veterinary rehabilitation practitioners to improve the security of foot placement by increasing nail traction. This study aimed to assess if nail grips would improve mobility outcomes in dogs recovering from knee surgery. The results did not show a significant effect on outcomes during the initial 2-week post-operative period; however, further research is needed to determine if they are useful for other applications.

**Abstract:**

The objective of this study was to assess the functional outcomes of dogs wearing nail grips in the first 2 weeks following tibial plateau leveling osteotomy (TPLO). Thirty dogs were included (*n* = 13 nail grips and *n* = 17 sham grips). Visual lameness scores (VLS), total pressure index (TPI), and client-specific outcome measures (CSOMs) were obtained by blinded observers on day 1 and day 14 +/− 3 post TPLO. CSOMs were also obtained on day 7. There were no differences in VLS and TPI between the treatment and sham group on day 14 (*p* = 0.44 and *p* = 0.59, respectively) or at any time point. CSOMs assessing walking on slippery flooring, ability to rise, and consistent use of surgical limb on a 5 min walk were also not different between groups (*p* = 0.78, *p* = 0.80, and *p* = 0.63) at any time point. Nail grips were well tolerated in dogs after orthopedic surgery. This study did not demonstrate a benefit for dogs wearing nail grips during the first two weeks after TPLO; however, further studies are warranted.

## 1. Introduction

Cranial cruciate ligament (CCL) rupture is one of the most common sources of hindlimb lameness in the canine patient [1,2]. Each year, billions of dollars are spent in the United States on the treatment of canine CCL ruptures in dogs [3]. Tibial plateau leveling osteotomy (TPLO) is one of the most common methods of surgical stabilization of the CCL deficient canine stifle [4,5,6,7]. Assistive devices could improve recovery and aid in the rehabilitation of dogs post TPLO surgery. However, evidence to support the use of these devices is lacking in current veterinary scientific literature.

Weight-bearing is crucial to maintaining joint health and bone density, delaying muscle atrophy, and preventing reinjury [8,9,10,11,12]. It has been shown that non-weight bearing results in decreased cartilage thickness and decreased proteoglycan content and synthesis [11,13,14]. Lack of weight bearing has been shown to slow woven bone formation in tibial defects in dogs [15]. Early mobilization has been demonstrated to reduce anterior knee pain, improve range of motion, and improve overall outcomes in human patients post anterior cruciate ligament reconstruction [16,17,18].

Nail grips (Dr. Buzby’s ToeGrips^®^, Beaufort, SC, USA) are rubber devices applied to the distal weight-bearing nails that purportedly enhance paw traction in dogs by increasing nail friction to grip the floor. Nail grips may be a useful tool to increase weight-bearing in post-operative patients by improving confidence in foot placement. A previous study has shown that nail grips increase stance time in the limbs of normal dogs which would be desirable in dogs recovering from orthopedic surgery [19]. In contrast to full coverage canine footwear, nail grips are designed for continuous wear, maintaining sensory feedback by direct paw-to-floor contact, and are less apt to trap moisture associated with pododermatitis. While widely used by rehabilitation practitioners for dogs with orthopedic disorders, evidence is lacking to support the efficacy of nail grips in improving weight-bearing and limb use. Dogs recovering from TPLO surgery are often reluctant to weight bear in the immediate post-operative period. In addition, CCL deficiency has been associated with decreased conscious proprioception [20,21,22] which may predispose to slipping. To date, there has been no research demonstrating how footwear could impact recovery post TPLO in the canine patient.

The objective of this study was to assess the functional outcomes of dogs wearing nail grips in the first 2 weeks following TPLO. The authors hypothesized that dogs wearing nail grips have significantly improved visual lameness scores, total pressure index, and client-specific outcome measures following TPLO compared to dogs wearing sham devices.

## 2. Materials and Methods

### 2.1. The Animals

Dogs recruited for enrollment were scheduled for TPLO surgery at the Schwarzman Animal Medical Center (AMC) in New York. The study was approved by the Institutional Animal Care and Use Committee of the AMC. The owners of the dogs provided written consent prior to enrollment. Dogs undergoing TPLO surgery were included in the study if they weighed over 7 kg and were ≥2 years old. Partial or complete cranial cruciate ligament ruptures were confirmed intraoperatively. Exclusion criteria consisted of dogs symptomatic for concurrent orthopedic diseases, comorbidities such as neoplasia or advanced metabolic disease, dermatological conditions affecting the nail bed, neurological deficits, or the use of epileptic drugs or systemic steroids.

### 2.2. Study Enrollment

Prior to enrollment, dogs had a complete physical and orthopedic exam by a board-certified surgeon, orthogonal radiographs of the affected stifle, and baseline CBC and biochemical profiles. Dogs eligible for the study from March 2018 to February 2020 were selected for enrollment two days prior to TPLO.

### 2.3. TPLO

After study enrollment, animals were admitted for surgery (day 0). Individual anesthetic protocols varied according to the anesthetist’s preference and individual needs. Stifle joints were evaluated using arthrotomy or arthroscopy at the surgeon’s discretion. TPLO was then performed as previously described [23] by a board-certified surgeon or a surgical resident under the direct supervision of a board-certified veterinary surgeon. If meniscal damage was detected, a partial meniscectomy was performed. 

### 2.4. Post-Operative Rehabilitation

All dogs received two standardized rehabilitation treatments (one immediately post-operative and one the next morning; Table 1). Therapy consisted of compressive cryotherapy (Game Ready Cold Equine, Cool Systems Inc., Berkeley, CA, USA), passive range of motion of the affected limb, low-level laser therapy, pulsed electromagnetic field therapy (PEMF) (Assisi Loop, Assisi Animal Health, Northvale, NJ, USA), transcutaneous electrical nerve stimulation (TENS) (TENS 7000, Middleburg Heights, OH, USA), and pulsed joint compression to the affected stifle as described elsewhere [24]. Therapies were performed by a certified rehabilitation therapist who varied based on availability.

### 2.5. Treatment Group Assignment and Treatment Application

Dogs were randomly assigned by statistical software (SAS plan procedure random number generator with seed ID 8675309, SAS Institute Inc., Cary, NC, USA) to one of two groups: one group received nail grips (Dr. Buzby’s ToeGrips^®^, Beaufort, South Carolina, USA) (NG; Figure 1A) applied with cyanoacrylate-based adhesive (Scotch^®^ Super Glue Liquid, St Paul, MN, USA) to the four weight-bearing nails of all four limbs within the first hour following TPLO surgery as described in a prior study [19]. The control group received a sham device (SG) also applied with cyanoacrylate-based adhesive following surgery. The sham nail grip was designed with a coved ventral surface such that the base (the primary mechanism of action) did not contact the floor and therefore provided no traction (Figure 1B). This design was created in collaboration with the manufacturer to mimic worn grips that have lost their efficacy. Grip size was assigned according to dog nail circumference as recommended by the manufacturer (Dr. Buzby’s ToeGrips^®^, Beaufort, SC, USA). The evaluator and investigator were both blinded and the person applying the nail grips was not involved in evaluating the patient. Clients were also blinded to group assignment and placebo grip design. Dogs were discharged from the hospital the day after surgery with 7 days of oral codeine (Codeine West-ward Pharmaceuticals Corp., Eatontown, NJ, USA), 14 days of trazodone (Trazodone, Teva Pharmaceuticals USA Inc., Parsippany, NJ, USA) and one type of nonsteroidal anti-inflammatory drug (NSAID) (Deramaxx, Elanco Animal Health, Greenfield, IN, USA; Galliprant, Elanco Animal Health, Greenfield, IN, USA; Previcox, Boehringer Ingelheim, Ingelheim am Rhein, Germany; Rimadyl, Zoetis Inc., Kalamazoo, MI, USA). Post-operative instructions were provided to clients that included restriction of activities to 5 min elimination walks and avoidance of stairs. Dogs were discharged with a sling to assist as needed with walking to prevent falls. Each client was provided home instructions for 5 min of passive range of motion of the hip, stifle, and tarsus of the surgical limb and hindlimb weight shifting toward the surgical limb to be performed for 10 repetitions twice daily. Owners were instructed to not perform weight-shifting exercises if their dog was not weight-bearing.

### 2.6. Outcome Measures

There were 3 outcome measures evaluated: visual lameness score [25], client-specific outcome measures [26,27], and total pressure index (Gait4Dog Walkway, Sparta, NJ, USA) (TPI) [28,29]. In addition, the number of grips lost throughout the study period was noted to account for the possible effect on outcomes. Study data were collected and managed using REDCap (REDCap (Research Electronic Data Capture) is a secure, web-based application designed to support data capture for research studies, providing: (1) an intuitive interface for validated data entry; (2) audit trails for tracking data manipulation and export procedures; (3) automated export procedures for seamless data downloads to common statistical packages; and (4) procedures for importing data from external sources) electronic data capture tool hosted at the Schwarzman AMC [30].

Visual lameness scores were evaluated on day 1 (one day following TPLO surgery) and 14 ± 3 post-surgery by a blinded board-certified sports medicine and rehabilitation specialist (LXA) in a 12-foot (3.7 m) long tiled hallway. Evaluations were either in person or via video recording depending on scheduling availability. Video recordings included a clip of each dog walking away from and toward the camera. A four-point lameness scale was used as previously described [25].

Objective gait analysis was collected in a quiet room using a pressure-sensing walkway (Gait4Dog Walkway, Sparta, NJ, USA) on days 1 and 14 ± 3 post-surgery. The mat was (4.3 × 2 m) and contained 16,128 encapsulated sensors. Calibration was performed by the manufacturer and raw data was captured by software designed to analyze canine gait. Each dog was walked across the length of the gait analysis mat until three successful walks per subject were obtained for TPI data collection as performed in a previous study [29]. TPI is defined as the sum of peak pressure values obtained from each activated sensor by a single paw during the stance phase [29]. Values are expressed as a percentage of body weight. Dogs were walked by a handler on a loose leash unassisted at a comfortable walking velocity (0.9–1.2 m/s). Walks were considered successful if dogs maintained a level head carriage, relaxed and steady walking speed, there was no pulling on the lead, and varied less than 5% in velocity. TPI of the surgical limb was included in the statistical analysis. If dogs were not weight-bearing, objective gait analysis could not be performed and TPI was recorded as 0%.

Client-specific outcome measures (CSOMs) were assigned at days 2, 7, and 14 ± 3 post-surgery always by the same owner [26,27]. Day 2 and 7 CSOMs were conducted over the phone. Owners were asked to score their dog’s ability to complete three unassisted activities on a scale from 0–4 as follows: 0, no problem; 1, mildly problematic; 2, moderately problematic; 3, severely problematic; 4, impossible. Activities included walking across a slippery floor using the surgical limb, getting up from resting using the surgical limb, and consistently using the surgical limb on a 5 min walk (Appendix A). These activities were most commonly reported to improve with the use of nail grips according to aggregated consumer feedback provided by the manufacturer.

### 2.7. Statistical Methods

Power analysis was calculated based on percent TPI as an objective measure. A repeated measures design with one between factor and one within factor had two groups demonstrating a 12% between groups and 50% within group difference suggesting a sample size of 56 subjects to achieve a 90% power when used with a 5% significance level. After the first 20 subjects were enrolled, a post hoc analysis using collected TPI data was performed. This suggested a sample size of 28 dogs to achieve a power of 80% at the 5% significance level.

Baseline descriptive statistics are presented as mean and standard deviation and median and range, as appropriate. Between treatment groups (NG/SG), analyses of baseline variables and clinical variables (nerve block and epidural administration, days until recheck, history of contralateral CCL tear, complete versus partial CCL tear, lameness duration, use of arthroscopy, age, body weight) were performed using analysis of variance (ANOVA). Non-parametric analyses related to ordinal response variables (visual lameness score, CSOMs, TPI, grip loss, meniscal injury, body condition score, and sex) were carried out using a Wilcoxon test for median differences modelled by the treatment group. Ordinal continuous dependent and independent variables were analyzed using ordinary least squares regression where error residuals were normally distributed. Analysis for proportions of categorical variables was performed using Chi-Square analysis. Randomization of dogs to cohort by a random number based upon an initial seed was carried out using the Plan Procedure. Additional exploratory and inferential analyses were carried out using commercially available statistical software (SAS plan procedure random number generator with seed ID 8675309, SAS Institute Inc., Cary, NC, USA) where an unadjusted *p* < 0.05 was deemed significant.

## 3. Results

### 3.1. Dogs Included 

Thirty-five dogs were initially enrolled in the study. One dog was excluded due to intolerance of nail grip application and one dog was excluded due to an inability to tolerate postoperative NSAIDs. Fourteen dogs were randomly assigned to the nail grips group (NG) and 19 dogs were assigned to the sham group (SG). One dog in the sham group was dropped from the study on day 1 due to decreased carpal extension in standing post-operatively. Two dogs were excluded due to deviations in the post-operative protocol (use of pulsed electromagnetic field therapy, laser, and cannabidiol in the other). Thus, 30 dogs were included in the final data analysis (*n* = 13 nail grips and *n* = 17 sham grips).

The majority of the dogs in the study were mixed-breed dogs (*n* = 12) and Labrador Retrievers (*n* = 7), followed by American Staffordshire Terriers (*n* = 3). There were one each of Golden Retriever, Polish Lowland Sheepdog, Rottweiler, Chihuahua, West Highland Terrier, English Bulldog, Shiba Inu, and Beagle. There were 16 spayed females (*n* = 5 NG, *n* = 11 SG), 13 neutered males (*n* = 7 NG, *n* = 6 SG), and one intact female (NG). Seven dogs were affected on the right hindlimb (*n* = 2 NG, *n* = 5 SG) and 23 dogs were affected on the left hindlimb (*n* = 11 NG, *n* = 12 SG).

The NG group had a mean age of 6.6 y ± 2.1 (range = 3 to 11 y), mean body condition (on a scale from 1–9) of 6.2 ± 0.9 (range = 5 to 8), and mean weight of 27.1 kg ± 7.8 (range = 7.6 to 36.3 kg). The SG group had a mean age of 6.9 y ± 2.9 (range = 2 to 11 y), mean body condition of 6.2 ± 1.3 (range = 4 to 8 years), and mean weight of 27.6 kg ± 10 (range = 10.3 to 43.3 kg). The mean duration of lameness in the NG group was 187.8± 137 days (range = 30–395 days) and 213.9 ± 262.9 days in the SG group (range = 30–1095 days). Five dogs in the NG group and 7 dogs in the SG group underwent arthroscopy. There were no differences in age, weight, sex, body condition score, and duration of lameness between SG and NG groups (*p* = 0.78, *p* = 0.89, *p* = 0.34, *p* = 0.35, *p* = 0.64, respectively). Incidence of meniscal injury, complete versus partial cranial cruciate ligament tear, and use of arthroscopy were also not significantly different between groups (Table 2). Eight more dogs in the SG group received epidurals (*p* = 0.03) and four more dogs in the NG group received femoral and sciatic nerve blocks (*p* = 0.02) compared to the NG group. Exactly the same number of dogs in each group received no regional analgesia (four per group). The mean elapsed number of days from surgery to recheck did not differ between NG and SG groups (13.4 ± 2.1 and 12.8 ± 2.2 respectively, *p* = 0.48). Oral post-operative medications included 1 mg/kg codeine (Codeine West-ward Pharmaceuticals Corp., Eatontown, NJ, USA) every eight hours and 5 mg/kg trazodone (Trazodone, Teva Pharmaceuticals USA Inc., Parsippany, NJ, USA) as needed. One dog received 2 mg/kg Deramaxx (Deramaxx, Elanco Animal Health, Greenfield, IN, USA) every 24 h, two dogs received 2 mg/kg Galliprant (Galliprant, Elanco Animal Health, Greenfield, IN, USA) every 24 h, one dog received 5 mg/kg Previcox (Previcox, Boehringer Ingelheim, Ingelheim am Rhein, Germany) every 24 h, and the remaining 26 dogs received 2.2 mg/kg Rimadyl (Rimadyl, Zoetis Inc., Kalamazoo, MI, USA) every 12 h.

There was no significant difference between NG and SG groups for visual lameness score, CSOMs(Table 3), and TPI (Table 4) at any time point. On day 14, median visual lameness scores were 1 ± 0.73 in the NG group and 1 ± 0.61 in the SG group (*p* = 0.62). Mean TPI on day 14 was slightly higher in the NG group (16.16 ± 1.62%) compared to the SG group (15.8 ± 1.87%), but these differences were not significant (*p* = 0.59). There was also no difference between the mean number of grips lost in the treatment group versus the sham group (4.38 and 3.76, *p* = 0.72).

### 3.2. Complications and Adverse Events 

Three dogs experienced diarrhea (1 NG and 2 SG dogs), one NG dog was reported to be foot licking, and one NG dog was observed to have mild interdigital erythema on the right manus. These clinical symptoms self-resolved without treatment. Decreased carpal extension in standing was observed in one NG dog which resolved once nail grips were removed.

### 3.3. Protocol Deviations

One dog in the SG group was discharged with a compression bandage that was removed the following day due to incisional hemorrhage. One dog in the SG group was excluded from the final data analysis due to starting a cannabidiol supplement during the study. Another dog in the SG group was excluded from the final data analysis due to unauthorized treatment with pulsed electromagnetic field therapy and therapeutic laser at home.

## 4. Discussion

This study evaluated the effects of nail grips on functional outcomes in dogs post TPLO. There was no significant difference between treatment and sham groups in lameness scores (VLS) or total pressure index (TPI) on days 1 and 14. Client-specific outcome measures (CSOMs) evaluating the ability to rise, walking on slippery floors, and consistent use of the surgical limb during a 5 min walk were also not significantly different on days 1, 7, and 14. Thus, our hypothesis that nail grips would improve functional outcomes was rejected. Nail grips were generally well tolerated and only one patient experienced an adverse event believed to be associated with grip placement.

There were several limitations in this study. Study design and/or selected patient population may have impacted the ability to show an effect of nail grips. Grips were applied on all feet, emulating shoe studies in people [31,32]. However, given the lack of studies on canine footwear for orthopedic disease, it is unknown if greater efficacy is achieved by grip placement on fewer limbs. While the sham model used in this study minimized the traction surface, it is possible that the placement of any device on a nail affects sensory input. This sensation could alter gait as was noted in the dog with decreased carpal extension excluded from the study. The addition of a no-grip group could have been considered; however, this would have increased bias by unblinded assessors. Lastly, since pain, range of motion, and weight-bearing are expected to rapidly improve in the first 2 weeks post TPLO compared to the immediate post-operative period [33,34], this may not have been the ideal patient population to note subtle differences between groups. Investigating other patient populations with chronic mobility disorders such as osteoarthritis or neurologic disease may be more successful at identifying the effects of nail grips.

It is possible that the standardized rehabilitation protocol accelerated recovery in both groups. The degree to which these post-operative therapies improved recovery is unknown as there is inconsistency in the literature reporting the benefits of modalities such as therapeutic lasers, TENS, and PEMF therapy [35]. However, as therapies were distributed equally among the groups, this should not have been a confounding factor. Realistically, there may have been some variation in owner compliance with home exercises. Weight shifting exercises were instructed only to be performed in weight-bearing dogs as without limb use, this exercise is ineffective at improving weight-bearing.

To the authors’ knowledge, there have been no other studies observing the effect of footwear on dogs with orthopedic disease. Increasing traction at the shoe-floor interface has been shown to prevent slips and falls in humans [36]. Slippery shoes decrease the coefficient of friction in humans and decrease body center of mass, coefficient of friction, acceleration, velocity, stride length, and tarsal dorsiflexion [31]. A human study comparing barefoot versus shod patients recovering from anterior cruciate ligament reconstruction demonstrated mildly increased knee extension during exercises when wearing shoes [32]. Conversely, other studies have demonstrated that increases in footwear traction may increase stress on the ACL during athletic activities by increasing tibial torque [37,38,39]. This suggests that post-operative patients may benefit from additional traction, whereas the same may be dangerous in a sports setting. Intuitively, dogs recovering from TPLO would benefit from improved traction; however, the ideal coefficient of friction in dogs has not been established. A study in normal dogs found that nail grips applied to all limbs increased stance time, decreased stride velocity, and decreased peak vertical force of the hindlimbs [19]. It was suggested that these effects on forces and gait may improve stability. More recently, a study assessing the kinetic effects of boots in normal dogs demonstrated an increase in peak vertical instantaneous loading rate [40]. However, further research is required to determine the effects in dogs with orthopedic or neurological mobility disorders.

There were several additional limitations of this study including low enrollment numbers, lack of additional outcomes collected, and variability of patient care. As pre-operative outcome measures were not collected, the effect of pre-existing lameness and function on this patient population is unknown. The authors elected to focus on clinically relevant functional outcome measures; however, thigh girth and goniometry may have also provided additional meaningful outcomes. While efforts were made to standardize intra and post-operative patient care, variability was unavoidable in a busy multi-doctor practice. Although one dog received a compression bandage, previous studies showed that Robert Jones bandages do not reduce post-operative swelling [41,42]. While a greater proportion of nail grip dogs received epidurals versus femoral and sciatic nerve blocks compared to the sham grip dogs, a previous study showed this difference should not impact time to ambulation [43] and there was no statistical difference between these groups.

## 5. Conclusions

The results of this study did not demonstrate the benefit of nail grips on visual lameness scores, total pressure index, or client-specific outcome measures in dogs during the first two weeks post TPLO. However, nail grips were generally well tolerated with minimal adverse effects. Further study investigating the use of nail grips and other footwear for canine mobility disorders is needed to determine if these products would be beneficial in improving rehabilitation efficacy.

## Figures and Tables

**Figure 1 animals-12-02312-f001:**
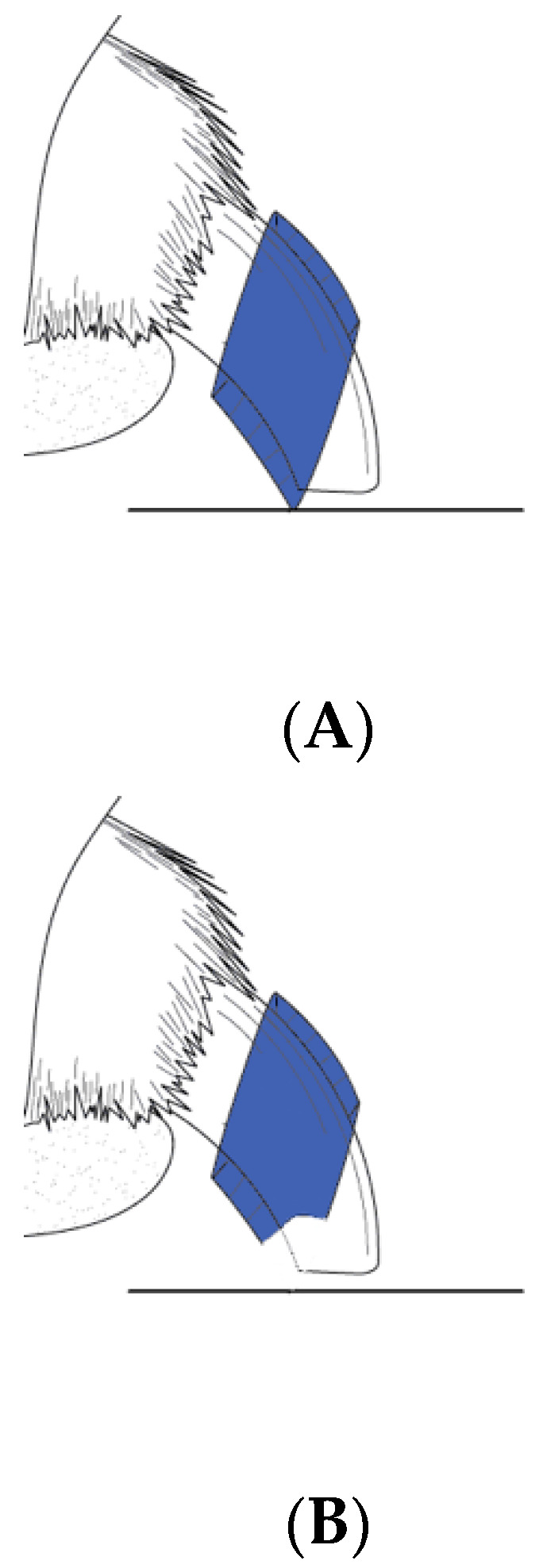
(**A**) Nail grip (Dr. Buzby’s ToeGrips^®^, Beaufort, SC, USA) with intact ventral grip surface contact. (**B**) Sham grip with coved ventral surface preventing floor surface contact.

**Table 1 animals-12-02312-t001:** Rehabilitation therapies provided to both groups to the surgical limb.

Therapy	Location	Specifications
Passive range of motion	tarsus, stifle, hip	10 repetitions
Pulsed joint compressions	stifle	10 repetitions
Pulsed Electromagnetic Field Therapy (Assisi Loop, Assisi Animal Health, Northvale, NJ, USA)	stifle (Circumferential)	15 min
Compressive cryotherapy (Game Ready Cold Equine, Cool Systems Inc., Berkeley, CA, USA)	stifle	20 min
Classe IIIb low-level laser therapy (Respond Systems 2400 XL, Respond Systems Inc., Branford, CT, USA)	stifle incision	1–2 J/cm^2^
	medial stifle joint	6–8 J/cm^2^
Transcutaneous Nerve Stimulation (TENS 7000, Middleburg Heights, OH, USA)	medial and lateral stifle	20 min; 100 Hz 50 μs

**Table 2 animals-12-02312-t002:** Relative incidence of complete (versus partial) cranial cruciate ligament rupture, epidural, arthroscopy, meniscal injury, and femoral/sciatic nerve block in nail grips and sham grips groups.

Variable	Treatment	Total	*p*-Value
Complete tear (versus partial)	Nail grips	8	0.28
	Sham grips	7	
Epidural	Nail grips	4	0.03
	Sham grips	12	
Arthroscopy	Nail grips	5	0.89
	Sham grips	7	
Meniscal injury	Nail grips	6	0.56
	Sham grips	6	
Femoral/sciatic nerve block	Nail grips	5	0.02
	Sham grips	1	

**Table 3 animals-12-02312-t003:** Median and ranges of visual lameness scores and client-specific outcome measures for the nail grips group as compared to the sham group.

Outcome	Day	Treatment	Median	Range	*p*-Value
Visual lameness score *	1	Nail grips	4	2–4	0.64
		Sham grips	4	2–4	
	14	Nail grips	1	0–3	0.44
		Sham grips	1	0–2	
Walking on slippery floor **	1	Nail grips	2	0–4	0.33
		Sham grips	1	0–4	
	7	Nail grips	0	0–3	0.64
		Sham grips	1	0–2	
	14	Nail grips	0	0–2	0.78
		Sham grips	0	0–2	
Rising using surgical limb **	1	Nail grips	2	1–4	0.63
		Sham grips	2	0–4	
	7	Nail grips	1	0–3	0.53
		Sham grips	1	0–3	
	14	Nail grips	1	0–3	0.80
		Sham grips	1	0–3	
Consistent use of surgical limb during 5 min walk **	1	Nail grips	4	1–4	0.24
		Sham grips	2	0–4	
	7	Nail grips	1	0–4	0.55
		Sham grips	1	0–2	
	14	Nail grips	0	0–2	0.63
		Sham grips	0	0–2	

* Visual lameness score criteria [25]: 0 = no observable lameness, 1 = intermittent, mild weight bearing lameness, 2 = moderate weight bearing lameness, 3 = severe weight bearing lameness, 4 = non-weight bearing. ** Client-specific outcome measures assessed patient difficulty completing 3 functional activities rated on a scale from 0 = no problem, 1 = mildly problematic, 2 = moderately problematic, 3 = severely problematic, 4 = impossible.

**Table 4 animals-12-02312-t004:** Mean ± SD total pressure index (TPI) for the nail grips group as compared to the sham group.

Day	Treatment	Mean	SD	*p*-Value
1	Nail grips	3.5	6.1	0.86
	Sham grips	3.1	5.9	
14	Nail grips	16.2	1.6	0.59
	Sham grips	15.8	1.9	

## Data Availability

Not applicable.

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
