# Peer review of "Effect of Nail Grips on Weight Bearing and Limb Function in 30 Dogs 2 Weeks Post Tibial Plateau Leveling Osteotomy"

_animals, 2022, doi:10.3390/ani12182312_

Round 1

Reviewer 1 Report

It is overall well presented. Here are a few comments: 

Line 21 and 23: please add “respectively” after p-values

Line 38: please provide page number for ref [5]. “Weight bearing is crucial to maintain joint health and bone density, delay muscle atrophy, and prevent reinjury[5]”

Line 77: please provide hospital location of the study.

Line 89: please clarify who performed rehab treatment.

Line 110: please clarify if patient only received one type of NSAID; perhaps “one type of nonsteroidal anti-inflammatories”, also please add abbreviation “NSAIDS”

Line 225: “1 dog received five mg/kg Previcox” please change to “5”

Line 246: “Lameness score criteria (Paschuck 2016)”. Please use reference #

Table 2: Lameness scorea: Please use Visual lameness score consistently throughout the study.

Table 2: Please use different superscript for: aLameness score criteria, and bClient specific outcome measures to prevent confusion with the footnotes. 

Line 281: Lameness score: Please use Visual lameness score consistently throughout the study.

Line 296: please provide ref for this statement: “Lastly, since most dogs rapidly improve in the first 2 weeks post TPLO”, and clarify improve in which areas, functional outcomes, ROM, weight bearing? 

Author Response

Dear Reviewer 1,

Thank you for taking the time to review our manuscript. We appreciate your comments and efforts to improve our work and hope these edits are satisfactory. 

Reviewer 2 Report

General comments:

This is a manucript about the use and effect of nail grips in the weight bearing ability in dogs after TPLO. It is a very interesting theme that is a main concern in rehabilitation of post-surgical dogs after TPLO. The manuscript need some important revisions before it is good for acceptance.

Specific comments:

Line 31: I suggest to replace "canine patients" for "dogs". Here and throughout the manuscript;

Line 30-36: I suggest to add some more bibliographic references in this paragraph. And the same throughout the introduction section. 

Line 57: In regard to the sentence "In addition, CCL deficiency ...". I do not understand, based on what neurophysiological mechanism? Could be the muscle weakness that originates the knuckling posture?  Or the fact that in TPLO, afferent pathways are affected, and also the ascending tracts to the somatosensory cortex, which give the information to the efferent pathways to correct foot position? Please review  this sentence and explain what you mean by that.

Line 69:  Was this study approved by an ethics committee ? If so please state this in the material and methods.

Line 69-75: Inclusion criteria should be more specific , including for example the lameness score ? Also they are acute, sub-acute or chronic dogs? These type of information should be described here.

Line 83: In regard to the TPLO surgery, I suggest to the authors to add the anesthetic and pharmacological protocol that was use (e.g. pre-medication, induction, maintenance ?) AINES were administered??

Line 92: Why was the laser class III chosen? maybe the authors could discuss this in the discussion section. Also here in the material and methods the authors should explain how was the laser treatment performed and the same in regard to TENS. How were the electrodes placed? Maybe the authors could add some pictures or a table describing the details, to make the manuscript more appealing and easy to read. 

Line 115: In regard to the sentence " Owners were instructed...". This should be mentioned and discussed in the discussion section.

Line 265: In regard to the complications and adverse effects mentioned , what was the medication performed and prescribed for their resolution. Please specify.

Discussion: In the discussion section it is where the authors need to make most of the changes. All discussion has to be approach in a different manner, with emphasis on the discussion of the results of the study. The authors need to discuss the results obtained and relate them with the bibliographic references.

Line 288: In this paragraph I suggest to the authors to discuss more clearly the outcomes of the study. It is important to understand and to discuss were the outcomes and results obtain , before and after the 15 days. 

Line 302: This paragraph should be developed and the authors should try to deepen this knowledge with human medicine references.

Line 223-225: I didn't really understood well this phrase " While a greater proportion ...". Please try to re-wrote for better understanding of the readers.

Conclusions: In the conclusion section, It is suggested for the authors to introduce the most important results obtained in the study. 

Bibliographic references: If possible, I think the authors should try to introduce some more updated references.

Author Response

Dear Reviewer 2,

Thank you for taking the time to review our manuscript. We appreciate your comments and efforts to improve our work and hope these edits are satisfactory. Your comments and feedback have helped this become a truly better paper. 

Round 2

Reviewer 2 Report

The authors follow through the main concerns of this manuscript, so for me it can be accepted for publication.